# Psychometric properties of the S-Scale: Assessing a psychological mindset that mediates the relationship between socioeconomic status and depression

**Julia Velten[1]\*, Saskia Scholten[1,2], Julia Brailovskaia[1], Jürgen Margraf[1]**

**1** Faculty of Psychology, Ruhr University Bochum, Mental Health Research and Treatment Center, Clinical Psychology and Psychotherapy, Bochum, Germany, **2** Universität Koblenz-Landau, Pain and Psychotherapy Research Lab, Landau in der Pfalz, Germany

\* julia.velten@rub.de

**Data Availability Statement:** All files are available from the Open Science Framework database (https://osf.io/s35yk/).

## Abstract

Individuals with low socioeconomic status (SES) are disproportionally affected by depressive disorders which are among the main causes for loss in healthy life years in adults worldwide. The main objective of the research presented here was to identify a psychological mindset of individuals with low SES and to investigate whether this mindset mediates the relationship between low SES and symptoms of depression. Towards these goals, a series of four studies was conducted: Study 1 identified a set of ten statements reflecting a psychological mindset associated with low SES using a population-based sample from Germany (N = 1,969). Study 2 cross-validated a psychometric scale (S-Scale) that was created based on these statements in a population-based sample from Germany (N = 3,907). Study 3 introduced a longitudinal perspective and showed that the S-Scale mediated the relationship between low SES and symptoms of depression assessed one year later in a German student sample (N = 1,275). Study 4 supported unidimensionality and construct validity of a unified version of the S-Scale and confirmed the mediation effect of the S-Scale for SES and depression while controlling for confounding variables (e.g., socially desirable responding) in a U.S. American convenience sample (N = 1,000). Evidence from four studies supported the reliability and validity of the S-Scale. Controlling for a psychological mindset as measured with this scale, low SES was no longer a predictor of depressive symptoms. The S-Scale can be used in clinical and research settings to assess a psychological mindset that puts individuals at risk for depression. Overall strengths of this series of studies include the use of population-based and longitudinal datasets and the application of findings to different operationalizations of SES. Future studies should investigate whether this mindset can be modified by psychological interventions and whether changes in this mindset predict improvements in depressive symptoms.

**Funding:** We gratefully acknowledge funding by the Alexander von Humboldt Professorship awarded to Jürgen Margraf by the Alexander von Humboldt-Foundation and by the Open Access Publication Funds of the Ruhr-Universität Bochum to Julia Velten. The funders had no role in study design, data collection and analysis, decision to publish, or preparation of the manuscript.

**Competing interests:** The authors have declared that no competing interests exist.

## Introduction

Depressive disorders are among of the most debilitating mental disorders and are the 6[th] leading cause for disability-adjusted life-years in adults globally [1]. Lifetime prevalence of major depressive episodes varies between countries with rates ranging from 3.6% in metropolitan China [2], and 9.9% in Germany, to 19.2% in the United States [3]. In addition, subthreshold or minor depression (i.e., clinically relevant depressive symptoms not meeting criteria for a major depressive disorder) is also associated with substantial economic costs [4], increased mortality [5], as well as a greater risk for developing a major depressive disorder [6]. According to the biopsychosocial model, depression is caused by interconnected biological, psychological, and social-environmental factors [7,8]. However, the relative contribution of these factors and the exact mechanisms by which depression is triggered and maintained have yet to be determined.

Over the past decades, researchers have become increasingly interested in neurobiological causes of depression. Despite the substantial amount of resources spent, biological correlates of depression such as dysfunction of the serotoninergic system [9] are only able to explain a fraction of variance in depression [10,11]. In contrast, societal factors such as socioeconomic status (SES) have received much less scientific attention despite striking evidence for the unequal distribution of depression between economically disadvantaged individuals and those who are better off [12,13].

### Low SES and depression

SES is an economic and sociological concept that has been described as "the position that an individual or family occupies with reference to the prevailing average of standards of cultural possessions, effective income, material possessions, and participation in group activity in the community" [Chapin, 1928, p. 99, cited by 14]. Variables used and combined to estimate SES are, e.g., years of education, family income, or possessions such as number of cars or computers per household. SES differences are found for rates of morbidity and mortality from almost every disease and condition with a linear gradient between SES and health [15]. Low SES is not only associated with increased rates of medical conditions such as diabetes [12] or cardiovascular disease [16], but also with a higher risk for mental disorders such as depression [12,13]. Unemployment is an indicator of low SES and its impact on depression was examined with meta-analytic methods across 130 studies. The overall effect size was $d = 0.50$ with unemployed individuals showing more symptoms of depression than employed individuals. A meta-analysis of 65 studies including, e.g., nationally representative epidemiological surveys and psychological autopsy studies showed not only a significant relationship between personal unsecured debt, an indicator of low SES, and depression (Odds ratio [OR] = 2.77), but also suicide completion (OR = 7.90), problem drinking (OR = 2.68), drug dependence (OR = 8.57), and psychotic disorders (OR = 4.03) [17].

The relationship between SES and depression is, at least to a certain extent, bidirectional. Living a life in relative or absolute poverty can increase the risk for developing depression (i.e., social causation) and functional impairment associated with depression (e.g., a lack of initiative) can limit the ability to, e.g., attain and keep a good paying job to secure one's SES (i.e., social drift). Previous studies suggest that mechanisms related to both social causation and social drift are relevant. Evidence for low SES being a cause of depression is especially robust [18]. Meta-analyses of longitudinal studies and natural experiments endorsed the assumption that unemployment is not only correlated to mental distress (including depression) but also causes it [19]. Both processes, however, are not exclusive and may be combined over the life cycle [20]. Support for the bidirectional relationship between SES and mental health comes,

e.g., from a community-based longitudinal study with 736 families in which low family SES was associated with several mental disorders in children (i.e., anxiety, depressive, disruptive, and personality disorders), while disruptive and substance use disorders in children were associated with poor education attainment [18].

## Conceptual framework of SES and health

To explain the mechanisms by which low SES can cause negative health outcomes, a conceptual framework has been developed [21], which proposes that SES has a direct impact on environmental resources and constraints (e.g., work situation, neighborhood factors). Support for the SES–environment association stems from empirical studies showing that individuals with low SES are more likely to have experienced difficult or stressful living situations such as unstable, crowded, or low-quality housing [22], unemployment [19] or jobs with less decision latitude [23]. The model also proposes that low SES has detrimental effects on several psychological variables. Together, both environmental and psychological factors are impacting health via access to medical care, exposure to pathogens, health-related behaviors, and chronic stress. While being informative concerning psychological factors as being mediators for the relationship of environment and somatic health, the framework does not a) differentiate between (relatively) stable psychological traits and mental disorders such as depression, and b) does not explain the effects of low SES on mental health.'

## Psychological mindset, low SES, and depression

A subset of the psychological factors listed by Adler and Steward [21] can be subsumed under the term *mindset* defined as "a mental attitude or inclination" or "a fixed state of mind" (Merriam-Webster, n.d.). Such a mindset has the power to control a person's attitude and influences a person's behavior. A *worldview* or weltanschauung describes "a comprehensive conception or apprehension of the world especially from a specific standpoint" (Merriam-Webster, n.d.). Individuals with low SES perceive the world, themselves, and their future in light of their previous experiences which significantly differ from those of individuals with higher SES in the above-mentioned aspects. With respect to Adler's and Steward's [21] conceptual framework, we propose that such a mindset is an important mediating factor between objective SES and mental health.

Support for the notion of a SES-specific mindset comes from studies on optimism and pessimism [24] defined as positive and negative generalized outcome expectancies [25]. In a population-based sample ($N$ = 9,711), indicators of low SES were associated with higher pessimism and lower optimism scores [26]. A study of 694 young adults showed that low SES in both childhood and adulthood was associated with dispositional optimism and pessimism, with childhood SES being more important for later pessimism than for optimism [27]. In a five-year longitudinal study of primary care patients ($N$ = 4,046), both low optimism and high pessimism at baseline were predictors of depression at follow-up in younger and middle-aged participants [28]. In two cross-sectional, representative samples of U.S. adults with more than 350,000 participants, low SES was associated with lower general life satisfaction [29] which is in turn related to depression [e.g., 30]. There is also evidence for individuals with lower SES experiencing less social support (i.e., feeling less loved and supported emotionally and receiving less tangible help from others) [31]. A meta-analysis showed consistent evidence for the protective role of social support against depression in children, adolescents, adults, and older adults [32]. Furthermore, in patients with major depression both the size of their social network and their subjective social support were predictive depressive symptoms at follow-up [33]. In a sample of 3,942 German seniors, resilience, the ability to bounce back in the face of

adversity [34], was higher among more educated individuals and those with higher household income [35]. Numerous studies with diverse samples such as immigrants to Russia and the United States [36], unemployed persons [37], or students at a Nigerian university [38] have consistently shown resilience to be associated with lower levels of depressive symptoms.

Without any claim to completeness, these traits are part of a psychological mindset of low optimism, high pessimism, low life satisfaction, low subjective social support, and low resilience that is characteristic for individuals with low SES. As each facet of this mindset has shown to be associated with depression, together, such a state of mind is likely to have detrimental effects on mental health. This is particularly dire as low SES has also been found to be associated with poorer outcomes of psychological therapies for depression [39], putting an even greater burden on the mental health of this vulnerable population.

## Aims of the present studies

So far, studies have focused on specific psychological traits and have examined such traits with respect to SES or depression without investigating their relevance as an important link between these two. Thus, the aims of the studies presented in the current paper were a) to identify a set of statements that reflect the psychological mindset of individuals with low SES, and b) to investigate whether this mindset mediates the relationship between low SES and symptoms of depression. Towards these goals, a series of studies was conducted using student, population-based, and convenience samples from Germany and the United States.

## The specific aims of the studies were as follows:

1. Study 1: To identify a set of items reflecting a psychological mindset associated with low SES using a population-based sample from Germany.

2. Study 2: To cross-validate a psychometric scale, entitled S-Scale, that was created based on the items of Study 1 in a different population-based sample.

3. Study 3: To introduce a longitudinal perspective. The hypothesis was that the S-Scale mediates the relationship between low SES and symptoms of depression assessed one year later.

4. Study 4: To assess dimensionality and construct validity of a unified version of the S-Scale. Further, we hypothesized that the mediation effect of the S-Scale for SES and depression would hold when controlling for several potential confounding variables (e.g., socially desirable responding).

Following, we will describe the four studies separately. For each study, detailed aims, participant characteristics, measures, data analysis, results, and summaries are provided. Hence, we will reflect on the overall results, strengths, and limitations in the discussion. All studies were carried out in accordance with the provisions of the World Medical Association Declaration of Helsinki (2013). The Ethics Committee of the Faculty of Psychology of the Ruhr University Bochum approved the studies (Reference number 073).

## Study 1

The goal of Study 1 was to identify a set of statements that reflect a psychological mindset of individuals with low subjective SES. Towards this goal, bivariate associations between individual items of five self-report questionnaires and subjective SES were examined. Subjective SES, the personal perception of one's rank or status within his or her society, is closely related to health variables [40–42], and can explain individual health and its changes above and beyond

objective SES [43]. Thus, it was selected as a reference point for identifying items reflecting a low SES mindset.

## Methods

**Procedure.** Study 1 included population-based data collected between November 2012 to February 2013 which were based on the register-assisted German census data from 2011 regarding age, gender, and education via systematized sampling procedures [for more information, see 30]. Participants had to be 18 years or older. No further inclusion or exclusion criteria were used. Data were gathered via online-surveys. All participants were informed about anonymity and voluntariness of the survey and gave written informed consent to participate. Participants did not receive a financial compensation.

**Participants.** Most participants ($N$ = 1,969; $M_{age}$ = 49.88, $SD$ = 18.21, *range*: 18 to 92; 51.5% female) indicated between 10 and 12 ($n$ = 1,114, 56.6%) or 13 years ($n$ = 405, 20.6%) of school education (see Table 1). Additional 21.6% ($n$ = 426) reported holding a university degree. The remaining 1.2% ($n$ = 24) were still attending school or indicated having left school without a degree.

**Measures.** *Subjective SES*. Subjective SES was assessed via self-report as participants were asked to identify as members of the lower, working, lower middle, middle, upper middle, or upper social class.

*Item pool*. To identify a psychological mindset that differentiates between individuals with higher and lower subjective SES, a selection of questionnaires measuring life satisfaction, optimism and pessimism, resilience, social support, and subjective happiness included in the "Bochum Optimism and Mental Health (BOOM)"-Studies were examined. The BOOM-Studies are a large international longitudinal project that investigates risk and protective factors of mental health. Life satisfaction was assessed with the Satisfaction with Life Scale (SWLS) [44], a 5-item self-report questionnaire measuring the judgmental component of personal well-being. Items were rated on a 7-point Likert scale ranging from 1 (*strongly disagree*) to 7 (*strongly agree*). Optimism and pessimism were measured with the 10-item Revised Life Orientation Test (LOT–R) [45,46]. Items of the LOT-R were rated on a 5-point Likert scale ranging from 0 (*strongly agree)* to 4 (*strongly disagree*). To measure resilience, the 11-item German Resilience Scale 11 (RS-11) was used [47,48]. Items were rated on a 7-point Likert scale ranging from 1 (*disagree*) to 7 (*agree*). Subjective experienced or anticipated support from the social network was measured with the 14-item German Questionnaire Social Support (F-SozU) [49]. Items of the F-SozU were rated on a 5-point Likert-scale ranging from 1 (*not true*) to 5 (*true*). To assess happiness, the 4-item Subjective Happiness Scale (SHS) was used [50]. Items were rated on a 7-point Likert scale ranging from 1 to 7.

**Table 1. Participants characteristics for Studies 1, 2, and 3.**

| | Study 1 (N = 1,969) | Study 2 (N = 3,907) | Study 3 (N = 1,275) |
|---|---|---|---|
| **Sample** | **Population-based sample, Germany** | **Population-based sample, Germany** | **University students, Germany** |
| Age M(SD), range | 49.88 (18.21), 18 to 92 | 46.77 (18.86), 18 to 98 | 24.03 (4.68), 17 to 58 |
| % of female participants | 51.5 | 51.8 | 61.8 |
| Education, n (%) | | | |
| 10 to 12 years | 1,114 (56.6) | 2,337 (59.8) | |
| 13 years | 405 (20.6) | 755 (19.3) | |
| University degree | 426 (21.6) | 793 (20.3) | |
| Other | 24 (1.2) | | |

**Data analysis.** To identify items most strongly associated with SES, absolute non-parametric partial correlations (Spearman's Rho) between items of the SWLS, LOT-R, RS-11, F-SozU, SHS and subjective SES controlling for age and gender were examined. To account for differences in scaling, all items were standardized. A maximum of ten items in total was set in order to create a short and concise scale that can be used in epidemiological studies and a maximum of three items per scale was set to prevent a potential overrepresentation of specific concepts.

## Results

**Subjective SES.** Thirty (1.5%) participants identified as members of the lowest social class, 228 (11.6%) as members of the working class, 337 (17.1%) as lower middle class, 1092 (55.5%) as middle class, 261 (13.3%) as upper middle class, and 21 (1.1%) as upper class. Subjective SES correlated positively with an objective measure of SES, namely household net income, $r$ (1681) = .37, $p < .001$.

**Extracted items.** Table 2 shows the ten items selected for the S-Scale. These items showed highest partial correlations to subjective SES. Content of the extracted items reflect a positive cognitive evaluation of one's life and living circumstances (e.g., SWLS, Item 1 and 2) as well as a pessimistic view of the world and one's future (e.g., LOT-R, Item 3 and 9). Further, items with highest correlations with SES indicated a perception of being able to cope with the challenges of life as well (e.g., RS-11, Item 5) as a general sense of happiness (e.g., SHS, Item 1).

**Table 2. Ten items with highest partial non-parametric correlations with perceived socioeconomic status controlled for age and gender ($N$ = 1929).**

| Original scale and item number | Item text | Range and rating scale | ρ with SES |
|---|---|---|---|
| SWLS, Item 2 | The conditions of my life are excellent | 1 (strongly disagree) to 7 (strongly agree) | .289* |
| SWLS, Item 1 | In most ways my life is close to my ideal. | 1 (strongly disagree) to 7 (strongly agree) | .187* |
| SWLS, Item 4 | So far, I have gotten the important things I want in life. | 1 (strongly disagree) to 7 (strongly agree) | .180* |
| SHS, Item 1 | In general, I consider myself . . . | 1 (not a very happy person) to 7 (a very happy person) | .168* |
| LOT-R, Item 3 | If something can go wrong for me, it will. | 0 (strongly agree) to 4 (strongly disagree) | .156* |
| LOT-R, Item 9 | I rarely count on good things happening to me. | 0 (strongly agree) to 4 (strongly disagree) | .147* |
| RS-11, Item 5 | I feel that I can handle many things at a time. | 1 (*disagree*) to 7 (*agree*) | .138* |
| LOT-R, Item 7 | I hardly ever expect things to go my way. | 0 (strongly agree) to 4 (strongly disagree) | .134* |
| RS-11, Item 3 | Keeping interested in things is important to me. | 1 (disagree) to 7 (agree) | .121* |
| RS-11, Item 9 | I can usually look at a situation in a number of ways. | 1 (disagree) to 7 (agree) | .105* |

* = $p < .001$, SWLS = Satisfaction with Life Scale [44], SHS = Subjective Happiness Scale [50], LOT-R = Life Orientation Test Revised [45,46], RS-11 = German Resilience Scale 11 [47,48].

## Summary

Ten items reflecting different psychological concepts (i.e., life satisfaction, happiness, optimism/pessimism, and resilience) were identified based on their partial correlations with subjective SES. Associations of selected items with subjective SES were small to medium in size.

## Study 2

The goal of Study 2 was to explore whether the items selected in Study 1 would differentiate between individuals with lower and higher objective SES and examine internal consistency of the newly developed S-Scale. In addition, we tested whether the S-Scale mediated the relationship between objective SES and symptoms of depression in a population-based sample. To provide first evidence for the incremental validity of the S-Scale, we compared the size of the mediation effect of the S-Scale to that of two other scales. Items of both scales were included in the new instrument (i.e., SWLS and LOT-R).

### Methods

**Procedure.** Like Study 1, Study 2 included population-based data based on the register-assisted German census data from 2011 regarding age, gender, and education via systematized sampling procedures. Data were collected between November 2012 to February 2013. [for more information, see 30]. The only inclusion criteria was age 18 years or older. No Exclusion criteria were used. Data were gathered via computer-assisted telephone interviewing. Trained professional interviewers conducted the telephone interviews with computer assistance. Participants were informed about anonymity and voluntariness of the survey and gave informed consent to participate verbally. A financial compensation was not paid.

**Participants.** In Study 2 ($N$ = 3,907; $M_{age}$ = 46.77, $SD$ = 18.86, $range$ = 18 to 89; 51.8% female), most participants indicated having attended between 10 and 12 years of school ($n$ = 2,337, 59.8%) and 19.3% ($n$ = 755) reported a high school degree after 13 years of education. Additional 20.3% ($n$ = 793) reported holding a university degree.

**Measures.** *SES.* As an objective measure of SES, a combined score comprising household income, occupation, and education of the main income earner in the household was calculated. Household income was assessed in nine categories ranging from less than € 750 per month (*Score 1*) to at least € 3500 per month (*Score 9*). Occupation was assessed with seven categories ranging from semi-skilled workers, students, and apprentices (*Score 2*) to Freelancers, senior civil servants, and managers (*Score 8*). Five categories were used to assess education with the lowest score reflecting 10 years of education or less (*Score 1*) and the highest score for holding a university degree (*Score 8*). A total score of 3 to 25 was calculated. For descriptive purposes, five different socioeconomic classes were examined: a score from 3 to 8 indicating the lowest class, 9 to 11 the second lowest class, 12 to 15 the middle class, 16 to 18 the second highest class, and a score of 19 to 25 indicating the highest socioeconomic class.

*S-Scale.* The ten items identified in Study 1 were combined and used to assess a low SES mindset. As item scaling was still based on their original questionnaires (e.g., SWLS, LOT-R), items were standardized and recoded in a way that higher values reflected a more *negative* mindset. Using these standardized and recoded items, internal consistency was good ($\alpha$ = .84).

*Depression.* Symptoms of depression were assessed with the 7-item depression subscale of the Depression Anxiety Stress Scales-21 (DASS-D) [51], a short version of the Depression Anxiety Stress Scales [52]. Items of the DASS-D were rated on a 4-point Likert scale from 0 (*did not apply to me at all*) to 3 (*applied to me very much, or most of the time*). Summing across the scale yields a total score ranging from 0 to 21. Cutoff-values of 5 and 7 were used to identify

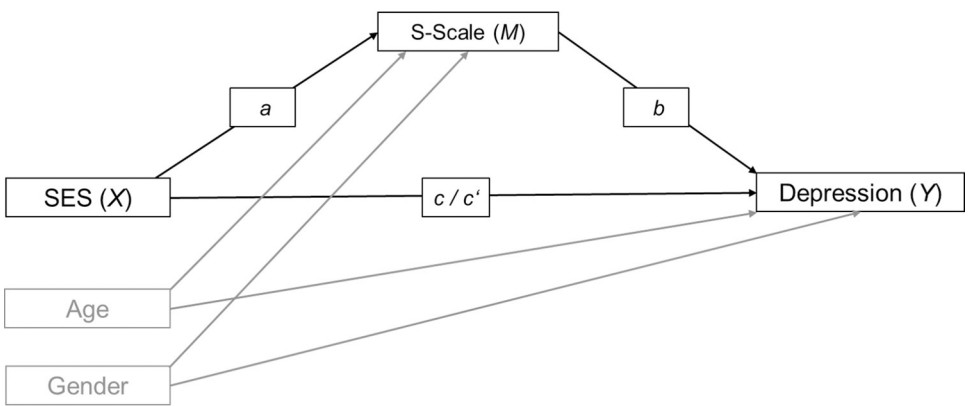

**Fig 1. Structure of mediation models.** Model with socioeconomic status (SES) as predictor (X). S-Scale as mediator (M), symptoms of depression as outcome (Y), as well as age and gender as control variables. Note. c = path of X to Y, without inclusion of M (total effect); a = path of X to M; b = path of M to Y; c' = path of X to Y including M (direct effect).

participants with mild or moderate symptoms of depression [52]. Internal consistency of the depression scale was excellent ($\alpha$ = .92).

**Data analysis.** Descriptive statistics were computed for age, gender, SES, S-Scale, and the DASS-D using SPSS 26. A mediation analysis using ordinary least squares path analysis was conducted to analyze the extent to which the effect of SES on depression was mediated through the psychological mindset as measured with the S-Scale (see Fig 1). For indirect effects, 95% percentile bootstrap confidence intervals (10,000 samples) are provided. Standardized regression coefficients as well as standardized effects for indirect effects are reported [53]. Mediation analyses were conducted using SPSS 26 in combination with the Process macro v3.5 [Process Model 4, 54]).

## Results

**Descriptive values.** The allocation of participants to the five socioeconomic groups ranging from lowest to highest was 3.4% ($n$ = 131), 16% ($n$ = 625), 37.3% ($n$ = 1,456), 17.9% ($n$ = 700), and 25.5% ($n$ = 996), respectively. Depression scores ranged from 0 to 21 ($M$ = 3.59, $SD$ = 4.28) with 8.9% ($n$ = 347) reporting mild and 21.5% ($n$ = 839) at least moderate depressive symptoms over the last week. Total score of the S-Scale ranged from -1.44 to 2.32 ($M$ = 0.00, $SD$ = 0.64, $Skewness$ = 0.39, $Kurtosis$ = -0.10) with higher values indicating a more negative mindset. The S-Scale showed significant differences across SES groups, $F(4, 3.90)$ = 83.80, $p < .001$, $\eta^2$ = 0.08. Post-hoc Scheffé-tests indicated that all SES groups significantly differed from each other (all $p$s < .037) with lower SES groups showing higher values on the S-Scale. Differences between the lowest SES group and other groups were large for the highest ($d$ = 1.11), second highest ($d$ = 0.93), and medium for the third highest ($d$ = 0.66), and small for the second lowest group ($d$ = 0.34).

**Mediation model.** Higher values on the S-Scale were associated with lower SES, $r(3806)$ = .28, $p < .001$, higher levels of depression, $r(3806)$ = .62, $p < .001$, and with being female, $r(3806)$ = -.05, $p < .001$. No significant associations with age were found, $r(3805)$ = -.02, $p$ = .226. Controlling for age and gender, SES was a significant negative predictor of depression ($c$ = -.12, $p < .001$; total effect) and of the S-Scale ($a$ = -.28, $p < .001$), as can be seen in Fig 2. Higher values of the S-Scale were associated with symptoms of depression ($b$ = .64, $p < .001$). A bootstrap confidence interval for the completely standardized indirect effect ($ab$ = -.18) was entirely below zero (-.20 to -.16) indicating that SES was indirectly associated with depression through its effect on a psychological mindset as measured with the S-Scale. The direct effect of

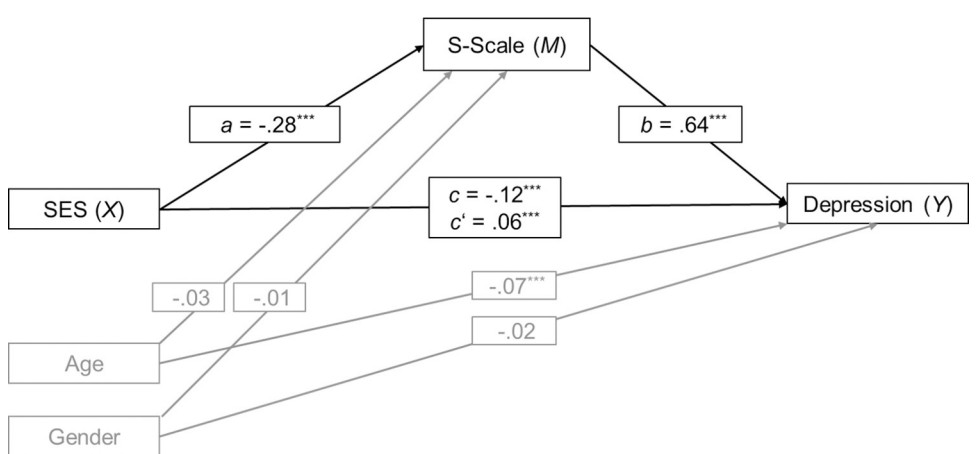

**Fig 2. Mediation model Study 2.** Standardized regression coefficients for the relationship between socioeconomic status (SES) and symptoms of depression as mediated by the S-Scale and controlled for age and gender. $^{***}p < .001$.

SES on depression independent of the S-Scale was positive ($c' = .06$, $p < .001$) suggesting that when controlling for this mindset, higher SES was actually related with *higher* levels of depression. Adding the S-Scale to the model of depression increased the amount of variance explained from $R^2 = .02$ to .39.

To assess the incremental validity of the S-Scale, two additional mediation analyses including either SWLS or LOT-R as mediating variables were conducted. For both scales, bootstrap confidence intervals for the completely standardized indirect effects (SWLS: $ab = -.13$; LOT-R: $ab = -.12$) were entirely below zero (SWLS: -.14 to -.11; LOT-R: -.14 to -.10). While both scales also mediated the relationship between SES and depression, the size of the effects was smaller than that found for the S-Scale.

### Summary

Study 2 showed, in a large population-based sample, that the psychological mindset measured with the S-Scale differentiated between individuals with lower and higher objective SES. Internal consistency of the scale was good. The S-Scale mediated the relationship between objective SES and symptoms of depression. The negative relationship between SES and depressive symptoms was reversed when the S-Scale was added to the model suggesting that when this psychological mindset is controlled for, higher SES can be associated with more symptoms.

## Study 3

Study 3 aimed to expand the findings of Study 2 by using a different conceptualization of objective SES and by introducing a longitudinal perspective. As other objective indicators such as household income might not adequately capture SES in student samples, we used a measure of SES in young individuals by taking the family situation during childhood and adolescence into account [55]. We hypothesized that the S-Scale would mediate the relationship between baseline SES and symptoms of depression assessed at 1-year follow-up even when baseline symptoms of depression were controlled for.

### Method

**Procedure.** Student data for Study 3 were collected from October 2017 to November 2018 at a German University [for more information, see 56]. Participants were recruited via email,

social media posts, and flyers. In addition to being enrolled as a student at the Ruhr University Bochum and 18 years or older, no inclusion or exclusion criteria were used. Data were gathered via online-surveys. Participants were informed about anonymity and voluntariness of the survey and gave written informed consent to participate. As compensation, participants could opt-in to receive course credits for their participation.

**Participants.** Participants ($N = 1,275$) were students at a large German university. On average, they were 24.03 years old ($SD = 4.68$, *range* = 17 to 58), and 61.8% were female.

**Measures.** *SES*. SES was assessed with the second version of the Family Affluence Scale (FAS) [55]. The FAS comprised four items asking about the number of cars and computers their family may have owned during the student's childhood, whether the student had had their own bedroom, and how often the student had traveled away on holiday with their family. For this study, a combined score ranging from 0 to 9 was calculated with higher values indicating higher SES.

*S-Scale*. The S-Scale as described in Study 2 was used. Internal consistency in Study 3 was acceptable with $\alpha = .77$.

*Depression*. As in Study 2, symptoms of depression were assessed with the DASS-D [51] that was administered both at baseline and at 1-year follow-up. Internal consistency of the DASS-D was good ($\alpha = .89$) at both assessment points.

**Data analysis.** Data were analyzed with the same strategy as described in Study 2. In the mediation model predicting depression at 1-year follow up, depression at baseline, age, and gender were included as covariates.

## Results

**Descriptive values.** Mean score on the FAS, used to assess SES in Study 3, was 4.62 ($SD = 1.72$, *range* = 1 to 9). Depression scores at both time-points ranged from 0 to 21 (baseline: $M = 4.37$, $SD = 4.45$; follow-up: $M = 4.40$, $SD = 4.49$) with 11.5% ($n = 145$) and 11.0% ($n = 138$) reporting mild and 24.3% ($n = 306$) and 24.5% ($n = 309$) at least moderate levels of depressive symptoms over the last week at baseline and follow-up, respectively. The S-Scale showed significant differences across SES groups based on quintiles of FAS scores, $F(4, 1269) = 10.99$, $p < .001$, $\eta^2 = 0.03$. Post-hoc Scheffé-tests indicated that the highest SES group significantly differed from the lowest three SES groups ($p \leq .003$, d $\geq 0.34$). Further, the second highest SES group differed significantly from the lowest SES group ($p = .003$, d = 0.38).

**Mediation model.** Higher values on the S-Scale were associated with lower SES, $r(1271) = -.17$, $p < .001$, higher levels of depression at baseline, $r(1259) = .53$, $p < .001$, and at 1-year follow-up, $r(1258) = .40$, $p < .001$, but were not significantly related to participants' age, $r(1273) = -.03$, $p = .325$, and gender, $r(1274) = .02$, $p = .566$. Depression at baseline and at 1-year follow-up showed a large positive correlation, $r(1258) = .60$, $p < .001$.

Controlling for age, gender, and baseline depression, SES was a significant negative predictor of depression at follow-up ($c = -.07$, $p = .004$; total effect) and of the S-Scale ($a = -.11$, $p < .001$), as can be seen in Fig 3. Higher values of the S-Scale were associated with symptoms of depression at follow-up ($b = .10$, $p < .001$). A bootstrap confidence interval for the completely standardized indirect effect ($ab = -.011$) was entirely below zero ($-.019$ to $-.004$) indicating that SES was indirectly associated with depression at follow-up through its effect on a psychological mindset as measured with the S-Scale. The direct effect of SES on depression independent of the S-Scale was slightly smaller than the total effect ($c' = .06$, $p = .017$). Adding the S-Scale to the model predicting depression at follow-up increased the amount of variance explained from $R^2 = .36$ to $.37$.

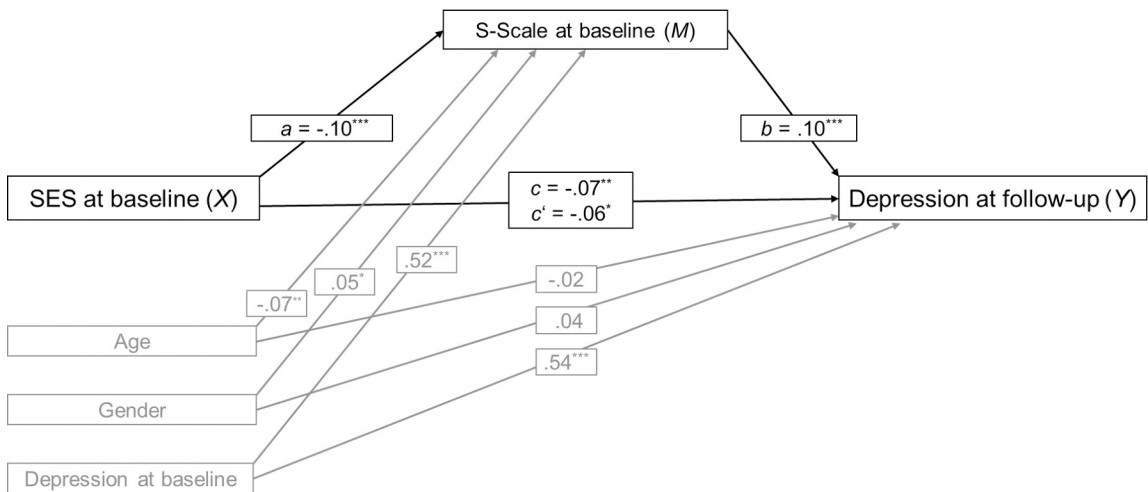

**Fig 3. Mediation model Study 3.** Standardized regression coefficients for the relationship between socioeconomic status (SES) and symptoms of depression at 1-year follow-up as mediated by the S-Scale and controlled for age, gender, and depression at baseline. $^{***}p < .001, ^{**}p < .01, ^{*}p < .05$.

## Summary

In Study 3, the S-Scale differentiated between university studies with low and high SES. It expanded previous findings by showing that the S-Scale mediated the relationship between objective SES and future symptoms of depression even when controlling for baseline levels of depression. Thus, individuals with a more negative mindset may be at risk for a deterioration of depressive symptoms.

## Study 4

The aim of Study 4 was to replicate the results of Study 2 with a unified, English version of the S-Scale in a sample of participants from the United States and to investigate dimensionality and construct validity of the S-Scale. Measures for convergent and discriminant validity were selected based on previous studies which showed associations (or a lack thereof) with depressive symptoms. Hypotheses for Study 4 were as follows: The scale was expected to be unidimensional and to show good internal consistency ($\alpha > .80$; Hypothesis 1).

To assess convergent validity, associations with self-esteem and experience of daily stressors, two variables with known associations with depression and mental health [57,58] were used. The S-Scale was expected to show moderate correlations to these two variables (Hypothesis 2). To assess discriminant validity, associations of the S-Scale and the psychological trait of narcissism was assessed. Narcissism is characterized by a grandiose sense of self, feelings of entitlement, and a dominant and antagonistic interpersonal style [59], which has been found only to show small correlations with depression [60,61]. Further, associations with socially desirable responding were expected to be moderate in size ($r < .5$). The S-Scale was also expected to mediate the relationship between objective SES and symptoms of depression when controlling for age, gender, and all of the above-mentioned potential confounding variables (i.e., self-esteem, daily stressors, narcissism, and socially desirable responding; Hypothesis 4).

## Methods

**Procedure.** Data for Study 4, comprising a convenience sample of participants located in the United States, were collected in February 2019 via Amazons Mechanical Turk (MTurk), an

online recruitment source. Prior research has indicated that participants recruited through MTurk provide valid data and are more demographically diverse than both U.S. university samples and standard Internet samples [62]. No inclusion or exclusion criteria were used except for the age being 18 years or older. Data were gathered via online surveys. All participants were informed about anonymity and voluntariness of the survey and gave informed consent to participate in written form. Participants received $ 2.70 financial compensation.

**Participants.** Participant characteristics for Study 4 are shown in Table 3.

## Measures

**SES.** Similar to Study 2, we calculated an objective measure of SES, comprising household income, occupation, and education of the main income earner in the household. Annual household income was assessed in twelve categories ranging from less than $ 10,000 to more than $ 159,000. Occupation of the main income earner was assessed with nine categories ranging from farm laborer/menial service workers to higher executive, proprietor of large businesses, and major professional (e.g., dentist, civil engineer). Twelve categories were used to assess education of the main income earner ranging from no schooling completed to a doctorate degree. A composite SES score was calculated as a mean of these variables.

**S-Scale.** In Study 4, a unified version of the S-Scale described in Study 2 was used. All items were presented with a 5-point Likert scale ranging from 0 (*does not apply to me at all*) to 4 (*does apply to me completely*). Higher values indicated a more negative psychological mindset (see S1 Table).

**Other measures.** Symptoms of depression were measured with DASS-D [51]. In Study 4, internal consistency of the DASS-D was excellent ($\alpha = .95$). Social desirability was assessed with a 12-item short version of the Balanced Inventory of Desirable Responding (BIDR) [63]. The BIDR is two-dimensional and assesses self-deceptive enhancement (honest but overly positive responding) and impression management (bias toward pleasing others). Items were rated on a scale from 1 (*very true*) to 7 (*not true*). A similar 16-item version of the BIDR has shown

**Table 3. Participant characteristics for Study 4.**

|  | Study 4 (n = 1,000) |
|---|---|
| Sample | Convenience sample, USA |
| Age M(SD), range | 37.27 (10.79), 18 to 74 |
| % of female participants | 47.0 |
| Education, n (%) |  |
| High school degree | 1,114 (56.6) |
| Some college credit but no degree | 405 (20.6) |
| Associate degree | 104 (10.4) |
| Bachelors degree | 451 (45.1) |
| Masters degree | 139 (13.9) |
| Doctorate | 15 (1.5) |
| Other | 34 (3.4) |
| Ethnicity, n (%) |  |
| White | 746 (74.6) |
| Black/African American | 102 (10.2) |
| Asian | 53 (5.3) |
| Hispanic | 52 (5.2) |
| Native American | 28 (2.8) |
| Other | 26 (2.6) |

good retest-reliability and acceptable validity [64]. In this study, internal consistency of the total scale was excellent ($\alpha$ = .90). Narcissism was assessed with the 13-item Narcissistic Personality Inventory-13 [59]. Items were rated in a forced-choice format (e.g., *I like having authority over people* vs. *I don't mind following orders*). The NPI-13 has demonstrated good convergent and discriminant validity and adequate overall reliability [59]. Internal consistency of the total scale was good ($\alpha$ = .83). To measure the participants' self-esteem, the Single-Item Self-Esteem Scale (SISE) was used [65]. Participants were asked to rate on a 5-point scale from 1 (*not at all true of me*) to 5 (*very true of me*) how much the statement "I have high self-esteem." applied to them. The SISE has been shown a reliable and valid alternative to longer measures of self-esteem [65]. The Brief Daily Stressors Screening Tool (BDSST) is a 9-item self-report scale that assesses the experience of general daily stressors in eight distinct life domains such as housing or employment [57]. It measures the subjective degree of stress on a 5-point Likert scale, ranging from 0 (*not at all*) to 4 (*very much*). In this study, internal consistency was good ($\alpha$ = .89).

**Data analysis.** Group comparisons were conducted as described in Study 2. To assess convergent and discriminant validity, bivariate correlations between scores of different questionnaires and the S-Scale were examined. In addition to Cronbach's $\alpha$, we employed an alternate internal consistency reliability estimation method, the McDonald coefficient-omega (coefficient-$\omega$), to calculate internal consistency reliability for scores for the unified S-Scale [66]. Confirmatory factor analysis was conducted to assess dimensionality of the scale, standardized factor loadings, and three fit indices were examined [67]: The comparative fit index (CFI) compares a hypothesized model's chi-square with that resulting from the independence model. A good model fit requires values above .95 [68]. The root mean square error of approximation (RMSEA) measures the difference between the reproduced covariance matrix and the population covariance matrix. with values less than .06 suggesting a small approximation error, indicating a good model fit and values between .08 and .10 a mediocre or acceptable fit. For the standardized root mean square residual (SRMR), values smaller than .08 indicated a good fit [69]. Owing to the large sample size, the $\chi$2-test was not interpreted. A mediation analysis was conducted with the same strategy as described in Study 2 but including age, gender, as well as daily stressors, self-esteem, narcissism, and socially desirable responding as covariates.

## Results

**Descriptive values.** Depression scores ranged from 0 to 21 (*M* = 5.93, *SD* = 6.26) with 12.2% (*n* = 122) reporting mild and 39.0% (*n* = 389) at least moderate levels of depressive symptoms over the last week. Total score of the unified S-Scale ranged from 0 to 3.60 (*M* = 1.49, *SD* = 0.57, *Skewness* = 0.20, *Kurtosis* = -2.85) with higher values indicating a more negative mindset. The S-Scale showed significant differences across SES groups based on quintiles of SES scores, $F(4, 995)$ = 7.29, $p < .001$, $\eta^2$ = 0.03. Post-hoc Scheffé-tests indicated that the lowest two SES groups significantly different from the highest SES quantile ($p \leq .002$, $d \geq 0.42$). Descriptive values of other variables were as follows: $M(SD)_{BIDR}$ = 51.11(16.25), $M(SD)_{NPI-13}$ = 6.23(1.43), $M(SD)_{SISE}$ = 3.36(1.18), and $M(SD)_{BDSST}$ = 12.52(8.63).

**Dimensionality and internal consistency.** Internal consistency of the unified S-Scale was good with $\alpha$ = .85 and $\omega$ = .83. A first, unidimensional model of the S-Scale showed an unacceptable fit via CFI = .603, TLI = .490; SRMR = .154, and RMSEA = .229, 90%CI [.220, .238]. After inspection of modification indices, error terms from items that were originally from the same scale were allowed to correlate. A unidimensional model of the S-Scale showed an acceptable to good fit via CFI = .951, TLI = .916; SRMR = .078, and RMSEA = .092, 90%CI [.082, .104]. Table 4 shows the standardized factor loadings of this model.

**Table 4. Standardized factor loadings.**

| Item number | Original scale and item number | *Standardized factor loadings* |
|---|---|---|
| 1 | SWLS, Item 2 | 0.59* |
| 2 | SWLS, Item 1 | 0.51* |
| 3 | SWLS, Item 4 | 0.53* |
| 4 | SHS, Item 1 | 0.65* |
| 5 | LOT-R, Item 3 | 0.69* |
| 6 | LOT-R, Item 9 | 0.38* |
| 7 | RS-11, Item 5 | 0.72* |
| 8 | LOT-R, Item 7 | 0.49* |
| 9 | RS-11, Item 3 | 0.28* |
| 10 | RS-11, Item 9 | 0.47* |

* $p < .001$, SWLS = Satisfaction with Life Scale [44], SHS = Subjective Happiness Scale [50], LOT-R = Life Orientation Test Revised [45,46], RS-11 = German Resilience Scale 11 [47,48].

Factor loadings ranged between .28 to .72 with an average of .53. The lowest factor loading was found for Item 9 (i.e., Item 9 from RS-11 "I can usually look at a situation in a number of ways."). When modeling the original scales as factors, fit indices were excellent, CFI = .988, TLI = .981; SRMR = .026, and RMSEA = .044, 90%CI [.033, .055].

**Convergent and discriminant validity.** The S-Scale showed a moderate positive correlation with the experience of general daily stressors, $r(997) = .49$, $p < .001$, as well as a moderate negative correlations with self-esteem, $r(997) = -.60$, $p < .001$. In addition, the S-Scale showed a small and non-significant correlation with narcissism, $r(997) = .05$, $p = .159$.

**Correlation and mediation analysis.** Bivariate correlations between SES, the S-Scale, depression, and the confounding variables controlled for in our mediation model are shown in Table 5.

Controlling for confounding variables, SES was not a predictor of depression ($c = -.008$, $p = .682$; total effect) and a negative predictor of the S-Scale ($a = -.11$, $p < .001$), as can be seen in Fig 4. Higher values of the S-Scale were associated with symptoms of depression ($b = .24$, $p < .001$). A bootstrap confidence interval for the completely standardized indirect effect ($ab = -.026$) was entirely below zero (-.039 to -.014) indicating that SES was indirectly associated

**Table 5. Bivariate correlations.**

| | SES | DEP | S-Scale | Age | Gender | Self-esteem | Daily stressors | Narcissism | Socially desirable responding |
|---|---|---|---|---|---|---|---|---|---|
| Socioeconomic status (SES) | 1 | -.04 | -.20*** | -.08* | .12*** | .21*** | -.03 | .08** | -.06 |
| Depressive symptoms (DEP) | | 1 | .61*** | -.20*** | .03 | -.37*** | .75*** | .14*** | -.63*** |
| S-Scale | | | 1 | -.12*** | .04 | -.60*** | .49*** | .13*** | -.45*** |
| Age | | | | 1 | -.14*** | -.02 | -.20*** | -.02 | .14*** |
| Gender | | | | | 1 | .08* | .00 | .10** | -.07* |
| Self-esteem | | | | | | 1 | -.22*** | -.05 | .32*** |
| Daily stressors | | | | | | | 1 | .14*** | -.61*** |
| Narcissism | | | | | | | | 1 | -.20*** |
| Socially desirable responding | | | | | | | | | 1 |

* $p < .05$

** $p < .01$

*** $p < .001$.

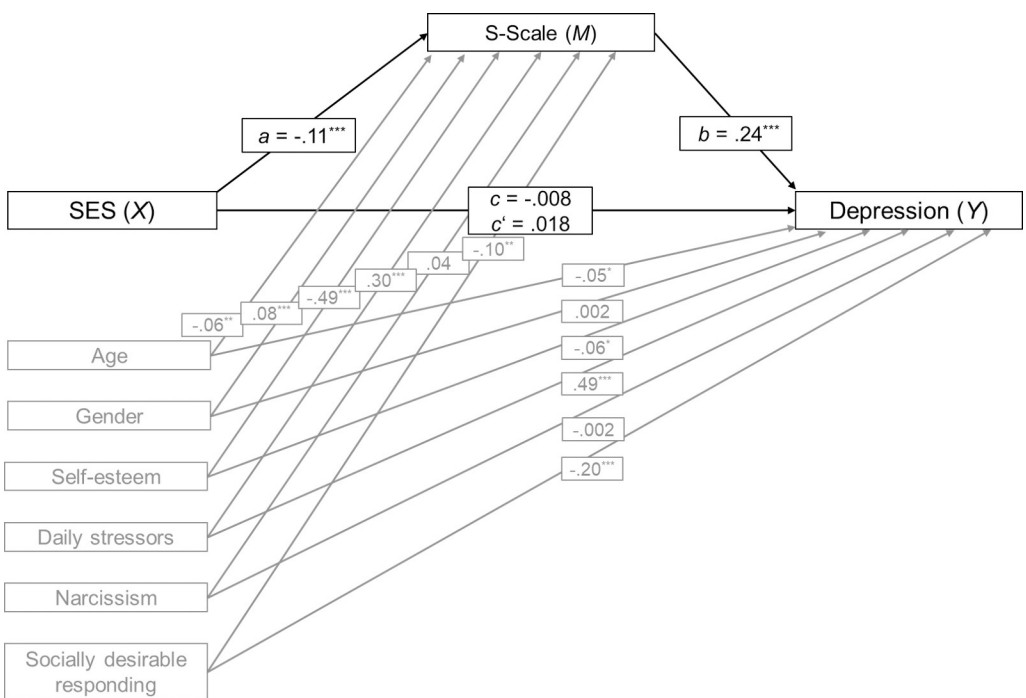

**Fig 4. Mediation model Study 4.** Standardized regression coefficients for the relationship between socioeconomic status (SES) and symptoms of depression as mediated by the S-Scale and controlled for age, gender, self-esteem, daily stressors, narcissism, and socially desirable responding ***$p < .001$, **$p < .01$.

with depression through its effect on the S-Scale. SES did not have a direct effect on depression independent of the S-Scale and confounding variables ($c'$ = -.008, $p$ = .682). Adding the S-Scale to the model increased the amount of variance explained from $R^2$ = .64 to .67.

## Summary

Study 4 supplemented previous studies by introducing a unified, English version of the scale and by providing further evidence for its usefulness in assessing a psychological mindset associated with low SES in a diverse sample of participants from the United States. The S-Scale Unidimensionality of the S-Scale was confirmed via acceptable to good fit indices. Two items, however, showed low factor loadings ($< .4$) revealing that these items have only weak influence on the latent variable. The S-Scale showed high internal consistency. It correlated with proximal variables in the expected directions and did not show substantial associations to the distal trait of narcissism. In a mediation model controlling for a variety of confounding variables, the mediation effect found in Studies 2 and 3 was replicated. The effect of SES on depression was fully attributable to the S-Scale and other confounding variables, no direct effect of SES on depression was observed.

## Discussion

We were able to identify a set of statements reflecting a psychological mindset associated with low SES. In a series of four studies, we confirmed that such a mindset differentiates between individuals with low and high SES and that it mediates the relationship between low SES and symptoms of depression in various populations. The S-Scale, created based on correlations of individual items of five different questionnaires with subjective SES, reflects a mindset

comprising low life satisfaction, low subjective happiness, low optimism, high pessimism, and low resilience. In incorporating these different aspects, the S-Scale offers a quick and easy measurement of this mindsets using items that show the greatest variability between individuals with different SES. While the original questionnaires on which the S-Scale was based include 44 items with different scales and scale ankers, the unified S-Scale has 10 items and a 5-point Likert scale.

By omitting items that did not differentiate between high and low SES, the S-Scale reflects a specific mindset that is related to SES rather than a comprehensive assessment of specific psychological concepts. (Example items showing weak associations with SES that were not selected for the S-Scale: *I can also bring myself to do things that I don't really want to do*, *It is important for me to be constantly busy*, *I often find something to laugh about*.). In Studies 2 to 4, the appropriateness of the selected items was supported as the S-Scale successfully discriminated between high and low SES groups independent of the operationalization of SES that was used. We were able to show that the assessed mindset is related to changes in depressive symptoms over a time-period of one year suggesting that individuals with a more negative mindset are at risk of becoming even more affected by depressive symptoms over time. In turn, individuals with a more positive mindset might have the capacity to overcome symptoms of depression by e.g., actively taking steps towards recovery.

## Clinical implications

Concerningly high rates of depression have been found in individuals of the lowest SES groups across countries worldwide [70,71]. Our studies suggest that for many individuals the experience of objective environmental constraints associated with low SES is related to the development of a psychological mindset that can negatively affect their mental health. When controlling for this mindset, objective SES is in many cases no longer a significant contributor to symptoms of depression. This speaks to the importance of psychological antecedents of depression and points to potential targets for clinical interventions. Cognitive behavioral therapy (CBT) is the most extensively researched form of psychotherapy for mental health issues [72] and can be considered a gold-standard treatment for and depression [73]. CBT-based methods such as cognitive restructuring aim to modify dysfunctional thinking patterns and replace negative or overgeneralized assumptions about the world or oneself with more helpful, appropriate, and optimistic ones. Patients with high values on the S-Scale—independent of whether they belong to a low SES group or not—feel that they cannot handle many things at a time, do not count on good things happening to them, or expect things to go wrong wherever they can. Identifying and targeting these maladaptive thoughts may be a promising means to improve depressive symptoms.

However, the S-Scale might also be relevant in relation to other therapeutic strategies: Augmenting CBT with second generation antidepressant medication might also have a positive impact on individuals with this mindset because (low strength) evidence exists that work function is improved which in turn might enhance a more positive future perspective [74]. Finally, considering the content of the items of the S-Scale low self-efficacy expectation and narrow focus are apparent. Therefore, experiential therapeutic approaches like Gestalttherapy might also be helpful because they immediately increase the experience of self-efficacy [75,76].

Administering the S-Scale before and after treatments of depression might shed light on whether the psychological mindset a) can be changed by cognitive methods and b) is a predictor of treatment success. Further, as the S-Scale predicted development of future symptoms of depression, it can also be used to identify individuals at risk for depression who might benefit from targeted prevention strategies [77].

## Limitations

Due to the nature of Study 1 (i.e., a population-based survey on traits and mental health), a limited range of questionnaires was administered. Thus, some of the psychological influences of the conceptual framework of SES and health by Adler and Stewart [21] were not explicitly covered (e.g., perceived discrimination, lack of control). Interestingly, while we did not include a questionnaire assessing a lack of control, some of the items related to pessimism included in the S-Scale nonetheless tap into this concept (*If something can go wrong for me, it will, I hardly ever expect things to go my way*). Although we controlled for some confounding variables, used longitudinal data, and diverse samples, this series of studies does not provide proof of the causal role of a psychological mindset for the relationship between SES and depression. Factors not assessed in these studies can be responsible for SES, a mindset as well as depressive symptoms (e.g., chronic physical/health conditions, access to healthcare) or can mediate the relationship between these variables (e.g., alcohol consumption [78]). Further, we cannot rule out the relationship between a negative mindset and low SES is bidirectional: A negative mindset might discourage individuals to pursue academic possibilities or to choose, e.g., less paying jobs. To address this shortcoming, future studies could include multiple assessment points as well as repeated measurements of SES.

Another limitation is that a new construct (i.e., SES-related psychological mindset), and a new measure, the S-Scale, are presented at the same time. Thus, the validity of the new scale could only be assessed approximately. Content validity was justified with correlational associations with self-reported SES. Future studies should include specific measures that address psychological factors related to low SES such as perceived discrimination to evaluate content validity. A comprehensive investigation of incremental validity was not conducted (i.e., whether the scale adds to the explanation of the relationship between SES and depression over and above other, more comprehensive measures) and should be included in future studies, too. Further, as both the mediator and outcome variable were assessed via self-report questionnaires, general answer tendencies (i.e., a negativity bias) may have artificially increased associations between mindset and depression. We created and validated the S-Scale in two Western, educated, industrialized, rich and democratic (WEIRD) countries. Thus, generalizability to other world regions cannot be determined. The usefulness of the S-Scale has been shown in samples of German and US-American adults. However, as a unified version of the S-Scale was only used in the US, a test of measurement invariance was not conducted.

While we were able to recruit diverse samples (e.g., concerning age, gender, ethnicity, SES) representative of the general population (Studies 1 and 2), individuals with very low SES were most likely underrepresented in our studies. To administer the S-Scale in low SES populations that are considered vulnerable to depression (e.g., marginalized groups such as refugees) might shed light on whether the effect of their difficult life circumstances on depression is also mediated by the S-Scale. Furthermore, data in the four studies were assessed between 2012 and 2019. This is a relatively long-time interval which might limit the comparability of the findings between the studies and their currency.

## Conclusion

In a series of four studies, initial evidence for the validity and reliability of the S-Scale was found. The scale taps into a psychological mindset that is characteristic of individuals with low SES and is associated with an increased risk for depressive symptoms. This psychological mindset mediated the relationship between low SES and depression. Thus, the present study adds to the conceptual framework on SES and health by specifying psychological trait factors

that function as a connecting link between environmental living circumstances and mental health problems.

## Supporting information

**S1 Table. S-Scale.** English version.
(DOC)

## Author Contributions

**Conceptualization:** Julia Velten, Saskia Scholten, Julia Brailovskaia, Jürgen Margraf.

**Data curation:** Julia Velten.

**Formal analysis:** Julia Velten.

**Funding acquisition:** Jürgen Margraf.

**Methodology:** Julia Velten, Saskia Scholten, Julia Brailovskaia, Jürgen Margraf.

**Project administration:** Julia Velten, Saskia Scholten.

**Supervision:** Jürgen Margraf.

**Validation:** Julia Velten.

**Writing – original draft:** Julia Velten, Saskia Scholten, Julia Brailovskaia, Jürgen Margraf.

**Writing – review & editing:** Julia Velten, Saskia Scholten, Julia Brailovskaia, Jürgen Margraf.

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
