## [Decision Letter · Decision Letter 0]

11 Feb 2021

PONE-D-20-39877

What is the psychological mindset that mediates the relationship between socioeconomic status and symptoms of depression?

PLOS ONE

Dear Dr. Velten,

Thank you for submitting your manuscript to PLOS ONE. After careful consideration, we feel that it has merit but does not fully meet PLOS ONE’s publication criteria as it currently stands. Therefore, we invite you to submit a revised version of the manuscript that addresses the points raised during the review process.

We look forward to receiving your revised manuscript.

Kind regards,

Stephan Doering, M.D.

Academic Editor

PLOS ONE

Journal Requirements:

2)  In your Methods section, please provide additional information about the participant recruitment method and the demographic details of your participants for all the studies reported, and especially for study 4. Please ensure you have provided sufficient details to replicate the analyses such as: a) the recruitment date range (month and year), b) a description of any inclusion/exclusion criteria that were applied to participant recruitment, c) a table of relevant demographic details, d) a statement as to whether your sample can be considered representative of a larger population, e) a description of how participants were recruited, and f) descriptions of where participants were recruited and where the research took place.

3) Please clarify in your ethics statement whether you received ethics approval from an ethics committee in the US to conduct study 4

4) Please note that according to our submission guidelines (http://journals.plos.org/plosone/s/submission-guidelines), outmoded terms and potentially stigmatizing labels should be changed to more current, acceptable terminology. For example: “Caucasian” should be changed to “white” or “of [Western] European descent” (as appropriate).

5) Please include captions for your Supporting Information files at the end of your manuscript, and update any in-text citations to match accordingly. Please see our Supporting Information guidelines for more information: http://journals.plos.org/plosone/s/supporting-information.

6)  We noticed you have some minor occurrence of overlapping text with the following previous publication(s), which needs to be addressed:

- https://link.springer.com/chapter/10.1007%2F978-3-319-72066-1_2

- https://eric.ed.gov/?id=EJ836788

- https://journals.sagepub.com/doi/10.1177/2158244015621113

- https://bmcpublichealth.biomedcentral.com/articles/10.1186/s12889-018-5526-2/

In your revision ensure you cite all your sources (including your own works), and quote or rephrase any duplicated text outside the methods section. Further consideration is dependent on these concerns being addressed.

Reviewers' comments:

Reviewer's Responses to Questions

**Comments to the Author**

1. Is the manuscript technically sound, and do the data support the conclusions?

Reviewer #1: Yes

Reviewer #2: Partly

2. Has the statistical analysis been performed appropriately and rigorously? 

Reviewer #1: Yes

Reviewer #2: Yes

3. Have the authors made all data underlying the findings in their manuscript fully available?

Reviewer #1: Yes

Reviewer #2: Yes

4. Is the manuscript presented in an intelligible fashion and written in standard English?

Reviewer #1: No

Reviewer #2: Yes

5. Review Comments to the Author

Reviewer #1: Dear authors,

I consider your work to be rather positive, although the fact that the manuscript presents several studies is unprecedented for me. On the other hand, this fact can also be a strength of the manuscript. The following text provides recommendations and comments divided into several homogeneous parts, which are focused on individual sections of the manuscript.

TITLE, ABSTRACT, KEYWORDS

When reading the title, it seems that the content of the manuscript is focused on a different topic than it actually is. In other words, the title does not reflect the content. The added value of the research is the validation of the tool. In my opinion, it would be appropriate to mention this fact in the title as well. If possible, please do not formulate the title in the form of a question.

In the abstract, there is still potential and space to present the added value of the study as a whole. You can also highlight who will benefit from the presented findings and what research gap has been filled by your study. Please, indicate the basic characteristics of the research sample (country, sample size).

The keywords should include the fact that your research focuses on validation, as well as the tools you have used.

INTRODUCTION

I have no serious comments on this section. I appreciate the quality of the authors' work they have done. I would like to note that, for example, alcohol consumption has a significant effect on the relationships examined in this study. In this sense, the attention can be drawn to the research of the author Jurgen Rehm. Also, from a socio-economic point of view, I would recommend the following study: https://www.mdpi.com/1660-4601/17/23/8853

At the end of the theoretical section, the authors provide information about the aims of individual studies. The orientation in the study as a whole would be facilitated, if the authors also provided the structure of the study in this part.

MATERIALS, METHODS AND RESULTS

I would like to apologize to the authors in advance for any criticism, but from my point of view, the section with methodology and results contains several shortcomings that I hope the authors will address or explain.

I would suggest the authors to summarize the methodological specifications of all four studies into one whole. The current form is confusing and partial methodological information appears throughout the study, which seems chaotic. In the individual surveys, it is necessary to define the characteristics of the research samples through their identification variables (gender, social status, education, marital status, etc.). It could be better to present the characteristics of the research samples in tables, it would be clearer (ideally all four studies in one table). I am also concerned about the fact that the individual data are in a relatively large time lapse, while the oldest data seems too old to me. But the authors can no longer do anything about it. I accept that the manuscript is divided into parts according to the presented studies, but I am not sure if this is the most appropriate solution. For example, the aims appear in duplicate, which seemed chaotic to me. When providing the ethics committee, please state its number.

I am aware of the fact that certain statistical procedures are currently used and accepted in various scientific fields, although their use is not always the most appropriate. The parametric correlation coefficient of Pearson's r is not very suitable in this case. A more suitable option would be the use of Kendall's or Spearman's correlation coefficient. Also, the rate of correlation is relatively low, please provide an author's interpretation of this result. Given the number of observations, I would suggest considering the use of the bootstrap method. In order not to repeat myself, please reconsider the use of the statistical methods or provide information that the assumptions for the application of the tests have been met. Honestly, I usually see different thresholds for the reliability indices of the data structure than those you have declared (especially for SRMR). According to the thresholds I consider relevant (SRMR < 0.05 (0.08), RMSEA < 0.06 (0.08)), your results would be questionable. Please provide an interpretation of statistical significance after Table 2. The results would be clearer and better understood if the outputs of the descriptive analyses were presented in tables. In my opinion, it could be beneficial to enrich the study in the form of the descriptive analysis that would include a classification of the identifying characteristics of the research sample (e.g. gender; SES, etc.). I would appreciate a greater scope for interpreting the revealed results.

DISCUSSION AND CONCLUSION

Please be more specific when comparing your results with the results of other authors. I also recommend enriching them and discussing more. I suggest to the authors that they be more careful when interpreting the "strength" of their outputs.

The implications are presented very narrowly. Although I agree with the statements, it would be appropriate to enrich this text with other approaches. For example: What is the attitude of the gestalt therapy, adjuvant therapy with antidepressants? Can there be a trend to expand the scale of the therapeutic repertory to different socio-demographic groups? Or it is necessary to focus on one approach, e.g. CBT?

I appreciate the conscientiously prepared limitations of the research.

I believe that the comments and recommendations will help to improve your manuscript. I keep my fingers crossed for the next process.

Reviewer #2: The manuscript deals with a highly relevant issue regarding one of the main antecedents of depression: socioeconomic status (SES). Authors propose that selected items from different scales (measuring constructs related to SES) can compose a new scale. In turn, this composed new scale is proposed to the “psychological mindset” precursor of depression.

Four studies were conducted to reach the main objectives: (a) identify the psychological mindset of individuals from low SES-groups and (b) determine if this mindset mediates the association between SES and depression.

I consider studies of this type relevant as when we do research linking SES to outcomes, SES is only a proxy of being comparatively not so good than other community members. To better understand what mediates the relationship SES-Health outcomes is crucial to target preventative actions and interventions.

Study 1 objective was to identify the items from 5 self-reported measures (of life satisfaction, optimism-pessimism, resilience, subjective experiences or anticipated support from the social network, and subjective happiness) with stronger correlations with a subjective-SES question. Selected items were 10 (r range = .28 to .12) from four of the measures.

Some issues needing further clarification and that may improve the manuscript:

- Approach to construct definition: why only the five constructs were considered first to creat S-scale? Just from availability from previous surveys?

- In the selection of the best 10 items: why ten in total? Why max. of three per scale? There were “good” items not selected because of these criteria? There were “bad” items included to complete 10? What does it mean that some scales provide three while other constructs provide none? I found the rationale for item selection vague.

- Most correlations are low (below .20). What implications does it have for item selection in the S-scale?

- Ethics: Do authors have/require authorization from original authors to use items from their scales in a new scale? Please discuss this and provide documentation if necessary.

Study 2 was designed to determine if the selected items discriminate well low and high-SES individuals (measured with a composite of objective measures) and test a mediation model (SES – S-scale scores – Depressive Symptoms).

Results show that five SES-groups (defined by authors) differ in scores from the S-scale with small-to-large effect sizes. Furthermore, the S-scale scores mediated the effect of SES on depressive symptoms

Some issues needing further clarification and that may improve the manuscript:

- Mediation: the direction of the effects: why SES could not be the antecedent of psychological mindset?

- How was the sampling design? It is representative of the German population? How was data collected? More details would be valuable.

- Incremental validity: S-scale score is better mediator over and above the original measures (four constructs represented in the S-scale)?

- Why do authors not do psychometrics using this large sample?

Study 3 repeats objectives of study 2, with changes in sampling (now are students) and the measure of SES (now a self-reported retrospective measure). Depression was assessed at two-time points, which configures a highlight of the manuscript.

Results show that SES was indirectly related to depressive symptoms through the psychological mindset, relevantly controlling for depression at baseline. The authors conclude that “individuals with a more negative mindset may be at risk for a deterioration of depressive symptoms” (lines 399-400).

Some issues needing further clarification and that may improve the manuscript:

- S-Scales showed SES differences based on quartiles: why quartiles are used now, and previously (S2) five categories created by researchers were used (instead of quartiles o quintiles)?

- Why were no controls on SES at T2 performed (i.e.: changes occurring to students living conditions that may alter past recollection)?

Study 4 was designed to replicate study 2 and perform psychometric exercises (validation), including dimensionality, convergent, and discriminant validity. Author claim for a one-factor solution.

Some issues needing further clarification and that may improve the manuscript:

- The rationale for selecting measures for convergent and discriminant validity seems short.

- Why was unidimensionality set to be confirmed? There are no other possible models (e.g.: two-dimensions) to compare? Why was unidimensionality only checked here? Some exploratory analyses were performed?

- A test of measurement invariance between USA-Germany seems to be needed in this study, especially to draw comparisons between countries.

- Most scholars would consider RMSEA and to some extent, SRMR values, below optimal. Please offer rationale about cut-off scores used on model fit. TLI should be reported.

- Why were errors between the same original measure allowed to correlate? Does it make theoretical sense? Was that determined in advance? Is it based on modification indices?

- I could not find Item loadings from the CFA nor errors.

Overall comments:

- The SES-measurement approach varies across studies but there is no overarching rationale about measurement issues regarding different methods. Moreover, arbitrary thresholds are used in different study, jeopardizing the credibility of results.

- Validation plan: psychometric and measurement issues are not well address (content validity, dimensionality, association with other variables). I suggest this be done across studies.

- Especially relevant in this case is incremental validity as items from other measures and constructs are being used.

- Concerning reliability, only internal consistency was provided (using alpha). Omegas and other types of reliability are valuable to report when introducing a new measure.

- Mediations models: justification for the direction of the effects should be more robust (e.g.: some specific mindset could make people report lower SES, and that could be the case with subjective measures, or make individuals have lower educational achievement and thus, less paid jobs).

- Some possible confounding variables traditionally studied in depression (also associated with SES) have not been analysed (such as chronic physical/health conditions, physical activity, loneliness, stress, access to healthcare). To control these variables is highly revelant when diverse samples in terms of (e.g.) age (18-90 y.o.), place of residency, are being used –since they are affected differentially from these factors. Please discuss implications.

- Also, the manuscript may benefit from a more in-depth literature review. For instance, Pepper and Nettle (2017) proposed the behavioral constellation of deprivation, identifying a set of behaviors observed in low-SES individuals, mostly associated with feelings of less personal control and more present-orientation. To some extent, those constructs should be part of the proposed initial pool of items the S-scale was drew from, and eventually would be desirable to be included in the validity plan, more concretely in the incremental validity plan.

6. PLOS authors have the option to publish the peer review history of their article (what does this mean?). If published, this will include your full peer review and any attached files.

Reviewer #1: No

Reviewer #2: No

---

## [Author Response · Author response to Decision Letter 0]

14 Apr 2021

Journal Requirements:

1) Please ensure that your manuscript meets PLOS ONE's style requirements, including those for file naming. The PLOS ONE style templates can be found at https://journals.plos.org/plosone/s/file?id=wjVg/PLOSOne_formatting_sample_main_body.pdf andhttps://journals.plos.org/plosone/s/file?id=ba62/PLOSOne_formatting_sample_title_authors_affiliations.pdf

We have carefully revised the manuscript to ensure that it meets PLOS ONE’s style requirements. 

2) In your Methods section, please provide additional information about the participant recruitment method and the demographic details of your participants for all the studies reported, and especially for study 4. Please ensure you have provided sufficient details to replicate the analyses such as: a) the recruitment date range (month and year), b) a description of any inclusion/exclusion criteria that were applied to participant recruitment, c) a table of relevant demographic details, d) a statement as to whether your sample can be considered representative of a larger population, e) a description of how participants were recruited, and f) descriptions of where participants were recruited and where the research took place.

We have revised the Methods section(s) to provide all relevant information. 

3) Please clarify in your ethics statement whether you received ethics approval from an ethics committee in the US to conduct study 4

The studies were approved by the ethics committee of the Ruhr University Bochum in Germany. Due to the anonymous nature of the studies, no separated ethics committee from the US was involved. 

4) Please note that according to our submission guidelines (http://journals.plos.org/plosone/s/submission-guidelines), outmoded terms and potentially stigmatizing labels should be changed to more current, acceptable terminology. For example: “Caucasian” should be changed to “white” or “of [Western] European descent” (as appropriate).

We have changed the term Caucasian to White. 

5) Please include captions for your Supporting Information files at the end of your manuscript, and update any in-text citations to match accordingly. Please see our Supporting Information guidelines for more information: http://journals.plos.org/plosone/s/supporting-information.

We have added a caption for the S1 Table at the end of the manuscript. 

6) We noticed you have some minor occurrence of overlapping text with the following previous publication(s), which needs to be addressed:

- https://link.springer.com/chapter/10.1007%2F978-3-319-72066-1_2

- https://eric.ed.gov/?id=EJ836788

- https://journals.sagepub.com/doi/10.1177/2158244015621113

- https://bmcpublichealth.biomedcentral.com/articles/10.1186/s12889-018-5526-2/

In your revision ensure you cite all your sources (including your own works), and quote or rephrase any duplicated text outside the methods section. Further consideration is dependent on these concerns being addressed.

We have carefully reviewed these files and have corrected any overlap or redundancies. 

 

Reviewer #1: 

1) TITLE, ABSTRACT, KEYWORDS

When reading the title, it seems that the content of the manuscript is focused on a different topic than it actually is. In other words, the title does not reflect the content. The added value of the research is the validation of the tool. In my opinion, it would be appropriate to mention this fact in the title as well. If possible, please do not formulate the title in the form of a question.

To address this, we have revised the title to appropriately reflect the content of the study. It now reads: Psychometric properties of the S-Scale: Assessing a psychological mindset that mediates the relationship of socioeconomic status and depression.

2) In the abstract, there is still potential and space to present the added value of the study as a whole. You can also highlight who will benefit from the presented findings and what research gap has been filled by your study. Please, indicate the basic characteristics of the research sample (country, sample size).

Thank you for this suggestion. We have revised the abstract to include this information. 

3) The keywords should include the fact that your research focuses on validation, as well as the tools you have used.

We have added the following key words: Validation, Satisfaction with Life Scale, Revised Life Orientation Test, German Resilience Scale 11, German Questionnaire Social Support, Subjective Happiness Scale

4) INTRODUCTION

I have no serious comments on this section. I appreciate the quality of the authors' work they have done. I would like to note that, for example, alcohol consumption has a significant effect on the relationships examined in this study. In this sense, the attention can be drawn to the research of the author Jurgen Rehm. Also, from a socio-economic point of view, I would recommend the following study: https://www.mdpi.com/1660-4601/17/23/8853

Thanks for this helpful suggestion. We have reviewed this interesting paper and now refer to it in our discussion section. 

5) At the end of the theoretical section, the authors provide information about the aims of individual studies. The orientation in the study as a whole would be facilitated, if the authors also provided the structure of the study in this part.

We have revised this section to include more information on the structure of the study. 

6) MATERIALS, METHODS AND RESULTS

I would like to apologize to the authors in advance for any criticism, but from my point of view, the section with methodology and results contains several shortcomings that I hope the authors will address or explain. I would suggest the authors to summarize the methodological specifications of all four studies into one whole. The current form is confusing and partial methodological information appears throughout the study, which seems chaotic. In the individual surveys, it is necessary to define the characteristics of the research samples through their identification variables (gender, social status, education, marital status, etc.). It could be better to present the characteristics of the research samples in tables, it would be clearer (ideally all four studies in one table). I am also concerned about the fact that the individual data are in a relatively large time lapse, while the oldest data seems too old to me. But the authors can no longer do anything about it. I accept that the manuscript is divided into parts according to the presented studies, but I am not sure if this is the most appropriate solution. For example, the aims appear in duplicate, which seemed chaotic to me. When providing the ethics committee, please state its number.

Considering your concern, we have restructured the manuscript to increase clarity. However, we decided to retain a certain amount of redundancy in order to increase readability and to orient the reader about the aims of the respective studies. We hope that you agree with our decision. 

It is true that the studies reported in this manuscript span a time period from 2012 to 2019. Considering your concern, we included this issue in the Limitation section of the revised manuscript. However, please note that as all studies focus on associations between variables, and are not designed to inform about absolute levels of, e.g., the S-Scale, we are convinced that validity and reliability of the findings are not negatively impacted by this. 

We have also revised the presentation of the demographic information and have now included two tables; one including information on Studies 1 to 3 (German studies) and a second to describe Study 4 (US-American Study). 

We have also included the number of the ethics proposal in the revised manuscript. 

7) I am aware of the fact that certain statistical procedures are currently used and accepted in various scientific fields, although their use is not always the most appropriate. The parametric correlation coefficient of Pearson's r is not very suitable in this case. A more suitable option would be the use of Kendall's or Spearman's correlation coefficient. Also, the rate of correlation is relatively low, please provide an author's interpretation of this result. Given the number of observations, I would suggest considering the use of the bootstrap method. In order not to repeat myself, please reconsider the use of the statistical methods or provide information that the assumptions for the application of the tests have been met. Honestly, I usually see different thresholds for the reliability indices of the data structure than those you have declared (especially for SRMR). According to the thresholds I consider relevant (SRMR < 0.05 (0.08), RMSEA < 0.06 (0.08)), your results would be questionable. 

To address this comment, we have revised the correlation analysis and changed the coefficient from Pearson’s to Spearman’s Rho. Notably, this change did not lead to any changes for the selected items. 

We agree with this reviewer that different thresholds for fit indices have been reported in the literature. Thus, we have provided specific references for the cutoffs used for the confirmatory factor analysis in the respective data analysis section. 

8) Please provide an interpretation of statistical significance after Table 2. The results would be clearer and better understood if the outputs of the descriptive analyses were presented in tables. In my opinion, it could be beneficial to enrich the study in the form of the descriptive analysis that would include a classification of the identifying characteristics of the research sample (e.g. gender; SES, etc.). I would appreciate a greater scope for interpreting the revealed results.

To address this and other comments, we have included two additional tables showing the descriptive characteristics of the study samples. 

We have also added an interpretation of the statistical significance to the Notes of Table 5. 

9) DISCUSSION AND CONCLUSION

Please be more specific when comparing your results with the results of other authors. I also recommend enriching them and discussing more. I suggest to the authors that they be more careful when interpreting the "strength" of their outputs.

To address this and other comments by both reviewers, we have thoroughly revised the discussion section and now offer more nuanced interpretations of the strengths of our studies. 

10) The implications are presented very narrowly. Although I agree with the statements, it would be appropriate to enrich this text with other approaches. For example: What is the attitude of the gestalt therapy, adjuvant therapy with antidepressants? Can there be a trend to expand the scale of the therapeutic repertory to different socio-demographic groups? Or it is necessary to focus on one approach, e.g. CBT?

To address this and other comments, we have carefully revised our discussion to describe the relevance of our findings for other areas of research and clinical practice. However, we have focused our clinical implications on therapeutic approaches that specifically focus on the modification of dysfunctional cognitions. 

11) I appreciate the conscientiously prepared limitations of the research.

We would like to thank this reviewer for this acknowledgment. 

Reviewer #2: 

1) The manuscript deals with a highly relevant issue regarding one of the main antecedents of depression: socioeconomic status (SES). Authors propose that selected items from different scales (measuring constructs related to SES) can compose a new scale. In turn, this composed new scale is proposed to the “psychological mindset” precursor of depression. Four studies were conducted to reach the main objectives: (a) identify the psychological mindset of individuals from low SES-groups and (b) determine if this mindset mediates the association between SES and depression. I consider studies of this type relevant as when we do research linking SES to outcomes, SES is only a proxy of being comparatively not so good than other community members. To better understand what mediates the relationship SES-Health outcomes is crucial to target preventative actions and interventions.

Study 1 objective was to identify the items from 5 self-reported measures (of life satisfaction, optimism-pessimism, resilience, subjective experiences or anticipated support from the social network, and subjective happiness) with stronger correlations with a subjective-SES question. Selected items were 10 (r range = .28 to .12) from four of the measures. Some issues needing further clarification and that may improve the manuscript:

Approach to construct definition: why only the five constructs were considered first to creat S-scale? Just from availability from previous surveys?

We have selected the questionnaires based on availability in the “Bochum Optimism and Mental Health (BOOM)”-Studies. The BOOM-Studies are a large international longitudinal project that investigates risk and protective factors of mental health. 

2) In the selection of the best 10 items: why ten in total? Why max. of three per scale? There were “good” items not selected because of these criteria? There were “bad” items included to complete 10? What does it mean that some scales provide three while other constructs provide none? I found the rationale for item selection vague.

We decided on a 10-item limit based on expert opinion and practical considerations. The rationale was to create a measure that is long enough to include several aspects of a psychological mindset associated with low SES but also short enough to allow this measure to be included in larger epidemiological surveys, to be time efficient and not too burdensome for the participants. Furthermore, due to its shortness the S-Scale can be used in clinical samples that often consist of patients with a limited attention span. 

Other, comparably short scales that we developed over the past few years (e.g., the Positive Mental Health Scale, doi: 10.1186/s40359-016-0111-x) were very frequently used and cited underlining the usefulness of such concise scales for research purposes. 

3) Most correlations are low (below .20). What implications does it have for item selection in the S-scale?

Notably, the human behavior and mental health are very complex constructs that are influenced by many different external and internal factors. This is reflected by our present findings. 

4) Ethics: Do authors have/require authorization from original authors to use items from their scales in a new scale? Please discuss this and provide documentation if necessary.

All scales are free to use for scientific purposes. We have included all relevant references in the manuscript and have added them to the supplemental material (i.e., the S-Scale) as well. 

5) Study 2 was designed to determine if the selected items discriminate well low and high-SES individuals (measured with a composite of objective measures) and test a mediation model (SES – S-scale scores – Depressive Symptoms). Results show that five SES-groups (defined by authors) differ in scores from the S-scale with small-to-large effect sizes. Furthermore, the S-scale scores mediated the effect of SES on depressive symptoms. Some issues needing further clarification and that may improve the manuscript: Mediation: the direction of the effects: why SES could not be the antecedent of psychological mindset?

While we agree that a psychological mindset can indeed influence how individuals perceive their SES, three out of four studies used somewhat objective measures that may be less susceptible to these kinds of biases. 

To address the possibility of bidirectional effects, we have added the following information to our limitations section: Further, we cannot rule out the relationship between a negative mindset and low SES is bidirectional: A negative mindset might discourage individuals to pursue academic possibilities and choose, e.g., less paying jobs. To address this shortcoming, future studies could include multiple assessment points as well as repeated measurements of SES.

6) How was the sampling design? It is representative of the German population? 

As described in the overview of studies section, both Studies 1 and 2 include population-based data which were based on the register-assisted German census data from 2011 regarding age, gender, and education via systematized sampling procedures. Thus, both samples were selected to be representative of the German population. 

7) How was data collected? More details would be valuable.

Data for Study 2 were gathered via computer-assisted telephone interviews conducted by trained, professional interviewers. We have added this information to the description of the procedure of each study. 

8) Incremental validity: S-scale score is better mediator over and above the original measures (four constructs represented in the S-scale)?

The main strength of the new S-Scale is that it allows a short and concise assessment of a low-SES mindset with 10 items versus 40 items that would need to be assessed if the original measures were used. Especially in large scale epidemiological surveys, every item counts considering time and monetary costs which is why we argue for the S-Scale also for pragmatic reasons. Furthermore, due to its shortness, the S-Scale can be used in clinical samples that often consist of patient with a limited attention span to prevent participants’ cognitive overload.

9) Why do authors not do psychometrics using this large sample?

Study 3 repeats objectives of study 2, with changes in sampling (now are students) and the measure of SES (now a self-reported retrospective measure). Depression was assessed at two-time points, which configures a highlight of the manuscript.

For Studies 1 to 3, the original scaling of the items was used. Thus, we decided to focus our reports of psychometric properties to Study 4, the first study in which a unified version of the S-Scale was first introduced. 

10) Results show that SES was indirectly related to depressive symptoms through the psychological mindset, relevantly controlling for depression at baseline. The authors conclude that “individuals with a more negative mindset may be at risk for a deterioration of depressive symptoms” (lines 399-400). Some issues needing further clarification and that may improve the manuscript:

S-Scales showed SES differences based on quartiles: why quartiles are used now, and previously (S2) five categories created by researchers were used (instead of quartiles o quintiles)?

One of the main strengths of Study 2 includes a comprehensive assessment of different aspects of objective SES (i.e., income, education, and occupation). Five SES groups were calculated based on a combination of these indices. The groups were derived by recommendations of the professional opinion research institutes. To increase comparability, we have revised our approach in Studies 3 and 4 and are now reporting quintiles for these studies. 

11) Why were no controls on SES at T2 performed (i.e.: changes occurring to students living conditions that may alter past recollection)?

The FAS used to assess SES in Study 3 is a retrospective assessment of the childhood living circumstances. It asks about whether the students had had their own bedroom, and how often the student had traveled away on holiday with their family during their childhood. Thus, values on the FAS are not expected to change. Owing to this, stable or trait variables were only included in the first wave of the longitudinal assessments. 

12) Study 4 was designed to replicate study 2 and perform psychometric exercises (validation), including dimensionality, convergent, and discriminant validity. Author claim for a one-factor solution. Some issues needing further clarification and that may improve the manuscript: The rationale for selecting measures for convergent and discriminant validity seems short.

We have revised this section to increase clarity. However, the measures used to assess convergent and discriminant validity were selected based on practical considerations and we have now added more information on additional steps suggested for validation of the S-Scale to our discussion section. 

13) Why was unidimensionality set to be confirmed? There are no other possible models (e.g.: two-dimensions) to compare? Why was unidimensionality only checked here? Some exploratory analyses were performed?

We decided to focus on the examination of unidimensionality to test whether the assumption that the S-Scale captures a general “mindset” is reasonable. As the items were selected from different scales, assessing different psychological traits, it was expected that a two- or more-factor solution would provide a better fit. However, our analysis focused on whether it is fair to assume that these items can be seen as one “mindset” instead of providing the best-fitting multi-factorial model. 

14) A test of measurement invariance between USA-Germany seems to be needed in this study, especially to draw comparisons between countries.

We agree that this would be a very important next step and have included this in our discussion section. However, as Study 4 was the only study that included the unified version of the scale (vs. items that were rated according to their original scales), such a test was not feasible in this set of studies. 

15) Most scholars would consider RMSEA and to some extent, SRMR values, below optimal. Please offer rationale about cut-off scores used on model fit. TLI should be reported.

We have rephrased this section to clarify that some of the fit indices can be considered acceptable if not good or optimal. We have also included the TLI in our revised manuscript. References for the selected cut-off values are also included in the Data Analysis section. 

16) Why were errors between the same original measure allowed to correlate? Does it make theoretical sense? Was that determined in advance? Is it based on modification indices?

This decision was made both on the basis of theoretical assumptions as well as inspection of modification indices. It was assumed that the error terms of the items reflect shared variance of the respective underlying constructs and thus, we decided to account for that in our model. 

17) I could not find Item loadings from the CFA nor errors.

We have discussed this among the co-authors and have decided to include standardized factor loadings in the revised manuscript. 

18) The SES-measurement approach varies across studies but there is no overarching rationale about measurement issues regarding different methods. Moreover, arbitrary thresholds are used in different study, jeopardizing the credibility of results.

While we agree that the operationalization of SES differs between studies, we believe that constitutes a strength rather that a weakness of this manuscript. There is no agreed upon definition on what exactly constitutes SES in different populations and/or countries. For example, level of education might affect living circumstances differently in the US vs. Germany. Further, in adolescent or young adult populations (e.g., university students) measurement of SES based on income or education is not possible. 

Whenever there were established cutoffs or categories, we decided to use them in our studies (e.g., Studies 1 and 2). In the remaining studies, we revised the data analysis to compare quintiles rather than quartiles in order to increase comparability. 

19) Validation plan: psychometric and measurement issues are not well address (content validity, dimensionality, association with other variables). I suggest this be done across studies.

The main reason for limiting the psychometric analysis to Study 4 is that the remaining studies did not use a unified version of the S-Scale but the item format and scaling of the original scales. Thus, the results obtained for, e.g., construct validity, would not be applicable to the final scale as used in Study 4. However, we have elaborated on this shortcoming more in our revised discussion. 

20) Especially relevant in this case is incremental validity as items from other measures and constructs are being used.

We have added a discussion of this to our limitations section. However, the goal of this series of studies was not to create a comprehensive and all-compassing measure that adds to the explanation of the relationship between SES and depression above and beyond other measures but rather to create a short and concise assessment that can be used whenever time and resources do not allow the use of several longer questionnaires. 

21) Concerning reliability, only internal consistency was provided (using alpha). Omegas and other types of reliability are valuable to report when introducing a new measure.

To address this comment, we have added Mc Donald’s Omega to the psychometric properties reported for the S-Scale in Study 4. 

22) Mediations models: justification for the direction of the effects should be more robust (e.g.: some specific mindset could make people report lower SES, and that could be the case with subjective measures, or make individuals have lower educational achievement and thus, less paid jobs).

While we agree that a psychological mindset can indeed influence how individuals perceive their SES, three out of four studies used somewhat objective measures that may be less susceptible to these kinds of biases. 

To address the possibility of bidirectional effects, we have added the following information to our limitations section: Further, we cannot rule out the relationship between a negative mindset and low SES is bidirectional: A negative mindset might discourage individuals to pursue academic possibilities and choose, e.g., less paying jobs. To address this shortcoming, future studies could include multiple assessment points as well as repeated measurements of SES.

23) Some possible confounding variables traditionally studied in depression (also associated with SES) have not been analysed (such as chronic physical/health conditions, physical activity, loneliness, stress, access to healthcare). To control these variables is highly revelant when diverse samples in terms of (e.g.) age (18-90 y.o.), place of residency, are being used –since they are affected differentially from these factors. Please discuss implications.

We agree and have included a discussion of this to our limitations section. 

24) Also, the manuscript may benefit from a more in-depth literature review. For instance, Pepper and Nettle (2017) proposed the behavioral constellation of deprivation, identifying a set of behaviors observed in low-SES individuals, mostly associated with feelings of less personal control and more present-orientation. To some extent, those constructs should be part of the proposed initial pool of items the S-scale was drew from, and eventually would be desirable to be included in the validity plan, more concretely in the incremental validity plan.

Thanks for pointing out this very interesting publication. We have included it in the discussion and emphasized the need for additional validation of the scale.

---

## [Decision Letter · Decision Letter 1]

10 Jun 2021

PONE-D-20-39877R1

Psychometric properties of the S-Scale: Assessing a psychological mindset that mediates the relationship between socioeconomic status and depression.

PLOS ONE

Dear Dr. Velten,

Thank you for submitting your manuscript to PLOS ONE. After careful consideration, we feel that it has merit but does not fully meet PLOS ONE’s publication criteria as it currently stands. Therefore, we invite you to submit a revised version of the manuscript that addresses the points raised during the review process.

We look forward to receiving your revised manuscript.

Kind regards,

Stephan Doering, M.D.

Academic Editor

PLOS ONE

Journal Requirements:

Reviewers' comments:

Reviewer's Responses to Questions

**Comments to the Author**

1. If the authors have adequately addressed your comments raised in a previous round of review and you feel that this manuscript is now acceptable for publication, you may indicate that here to bypass the “Comments to the Author” section, enter your conflict of interest statement in the “Confidential to Editor” section, and submit your "Accept" recommendation.

Reviewer #1: All comments have been addressed

Reviewer #2: (No Response)

2. Is the manuscript technically sound, and do the data support the conclusions?

Reviewer #1: Yes

Reviewer #2: Partly

3. Has the statistical analysis been performed appropriately and rigorously? 

Reviewer #1: Yes

Reviewer #2: No

4. Have the authors made all data underlying the findings in their manuscript fully available?

Reviewer #1: Yes

Reviewer #2: Yes

5. Is the manuscript presented in an intelligible fashion and written in standard English?

Reviewer #1: Yes

Reviewer #2: (No Response)

6. Review Comments to the Author

Reviewer #1: Dear authors,

The issue analysed in the presented paper is clearly important from a social point of view and attractive for research.

I perceived the quality of the study at the first reading, but certain aspects worried me in terms of methodology and discussion. At this point, I would like to appreciate the work on improving the study. The manuscript was revised to a high standard and all comments were incorporated or explained. I would like to write that the current version is methodologically correct and the discussion has been reworked to the required level. The clarity of the study also increased.

I thank the authors for accepting and incorporating the comments and I wish them all the best.

Reviewer #2: (No Response)

7. PLOS authors have the option to publish the peer review history of their article (what does this mean?). If published, this will include your full peer review and any attached files.

Reviewer #1: No

Reviewer #2: No

---

## [Author Response · Author response to Decision Letter 1]

2 Jul 2021

Journal Requirements:

We have checked the references list and were not able to identify any retracted papers. We would greatly appreciate it if you could let us know whether we have overlooked anything. 

 

Editorial comments: 

1. I appreciate the authors for their responses. The revised version of the manuscript addresses many of the issues reviewers’ raised, and the manuscript is now more robust and looks balanced in its strengths and weaknesses. However, the main concerns were addressed superficially. New measures should improve previous measures in some way or be based on a new construct. The proposed S-scale somehow combines both: it tries to combine previous measures to create a new measure and proposes a new construct (“SES-related psychological mindset”). To represent a significant contribution to the scientific and psychological literature, new measures should be based on different reliability and validity considerations. This manuscript offers evidence of internal consistency (only one type of reliability) and some evidence of validity. However, critical aspects of validity for the proposed uses of the scale, such as incremental validity and content validity, are not discussed. 

We appreciate the editors nuanced view of our paper and have addressed the remaining concerns carefully. We hope that by doing so, the revised manuscript finds the editors approval. While we agree that our manuscript does not include all analyses that may be useful for the purpose of introducing this new construct, we believe that – especially after the current revision – our manuscript is an important starting point to stimulate further research. 

To further address this issue, we have included the following information in our discussion: Another limitation is that a new construct (i.e., SES-related psychological mindset), and a new measure, the S-Scale, are presented at the same time. Thus, the validity of the new scale could only be assessed approximately. Content validity was justified with correlational associations with self-reported SES. Future studies should include specific measures that address psychological factors related to low SES such as perceived discrimination to evaluate content validity. A comprehensive investigation of incremental validity was not conducted (i.e., whether the scale adds to the explanation of the relationship between SES and depression over and above other, more comprehensive measures) and should be included in future studies, too.

2. Content validity issues are very relevant for a new measure trying to measure a new construct but only justified in terms of correlational association with self-reported SES. Furthermore, the content of the retained items is not discussed – how they relate to each other to sum up to the new latent variable. When addressing incremental validity, the main question is: is the S-Scale necessary? Authors refer that the S-scale is short and can be used in epidemiological surveys. However, the question remains. All the other measures can be shortened to create reduced versions, and some are already short (5 items or less). How much added variance is explained by mixing up items of different constructs and introducing the “psychological mindset/S-Scale” variable. This question is critical and of much interest both for readers and the success of the S-Scale. Does the S-scale explain more variance when exploring if it mediates the association between SES and depression than optimism and life satisfaction? The authors can answer this question using data from Studies 2 and 3. 

To provide further information on the content of the extracted items, we have added a short summary of the themes covered to the Results section of Study 1. 

To address the incremental validity of the S-Scale, we have used data from Study 2 for which the other scale scores were available. In this study, the indirect (mediation) effect of the S-Scale was larger (.18) than those of the Satisfaction with Life Scale (-.13) and of the revised Life Orientation Test (-.13) supporting the usefulness of the newly created scale. Thus, we have added the following information to Study 2: To assess the incremental validity of the S-Scale, two additional mediation analyses including either SWLS or LOT-R as mediating variables were conducted. For both scales, bootstrap confidence intervals for the completely standardized indirect effects (SWLS: ab = -.13; LOT-R: ab = -.12) were entirely below zero (SWLS: -.14 to -.11; LOT-R: -.14 to -.10). While both scales also mediated the relationship between SES and depression, the size of the effects was smaller than that found for the S-Scale. 

3. Cutoff criteria suggested for assessing fit indices still need clarification. More justification for the selected values should be provided. Hu & Bentler criteria, despite cited, recommends other thresholds. 

Thanks for pointing this out. We have corrected the cutoff of the SRMR value to .8 as suggested by Hu and Bentler. 

4. In line 545, the correlation between S-scale and narcissism is minimal (.05), but in table 5 the value is .13***. On the contrary, correlations between S-scale and self- esteem and daily stressors are coincident.

We apologize for this mistake and have corrected the statement in line 545 to match the correct data that were listed in the table. 

5. What does it mean that items originally from the same scale were allowed to correlate? Maybe the authors mean they correlated the errors terms? Fit indices of competing models should be provided: with and without correlations between error terms and also modeling original constructs as factors. If the fit of the unidimensionality model is better, more evidence for the utility of the S-scale will be provided.

Thanks for giving us the opportunity to clarify that the error terms from items that were originally from the same scale were allowed to correlate. 

We have added fit indices of a model with uncorrelated error terms to the manuscript. The section now reads: A first, unidimensional model of the S-Scale showed an unacceptable fit via CFI = .603, TLI = .490; SRMR = .154, and RMSEA = .229, 90%CI [.220, .238]. After inspection of modification indices, error terms from items that were originally from the same scale were allowed to correlate. A unidimensional model of the S-Scale showed an acceptable to good fit via CFI = .951, TLI = .916; SRMR = .078, and RMSEA = .092, 90%CI [.082, .104].

When modeling the original scales as factors, fit indices are excellent, CFI = .988, TLI = .981; SRMR = .026, and RMSEA = .044, 90%CI [.033, .055]. Please note that our analysis aimed to provide evidence for the adequateness of a unidimensional solution, not at finding the best fitting factor structure. However, we have added these values to the manuscript as well to provide readers with a complete picture of the data.

---

## [Editor Report · Decision Letter 2]

27 Sep 2021

Psychometric properties of the S-Scale: Assessing a psychological mindset that mediates the relationship between socioeconomic status and depression.

PONE-D-20-39877R2

Dear Dr. Velten,

We’re pleased to inform you that your manuscript has been judged scientifically suitable for publication and will be formally accepted for publication once it meets all outstanding technical requirements.

Kind regards,

Stephan Doering, M.D.

Academic Editor

PLOS ONE

---

## [Editor Report · Acceptance letter]

7 Oct 2021

PONE-D-20-39877R2 

Psychometric properties of the S-Scale: Assessing a psychological mindset that mediates the relationship between socioeconomic status and depression. 

Dear Dr. Velten:

I'm pleased to inform you that your manuscript has been deemed suitable for publication in PLOS ONE. Congratulations! Your manuscript is now with our production department. 

Kind regards, 

on behalf of

Professor Stephan Doering 

Academic Editor

PLOS ONE